# Psychobiotics-like Activity as a Novel Treatment against Dry Scalp Related-White Flakes Production with *Pogostemon cablin* Leaf Extract

Marie Meunier [1,*], Emilie Chapuis [1], Cyrille Jarrin [2], Julia Brooks [3], Heather Carolan [3], Jean Tiguemounine [4], Carole Lambert [2], Bénédicte Sennelier-Portet [5], Catherine Zanchetta [2], Amandine Scandolera [1] and Romain Reynaud [2]

1 Givaudan Active Beauty, 51110 Pomacle, France; emilie.chapuis@givaudan.com (E.C.); amandine.scandolera@givaudan.com (A.S.)
2 Givaudan Active Beauty, 31400 Toulouse, France; cyrille.jarrin@givaudan.com (C.J.); carole.lambert@givaudan.com (C.L.); catherine.zanchetta@givaudan.com (C.Z.); romain.reynaud@givaudan.com (R.R.)
3 Givaudan UK, Ashford TN24 0LT, UK; julia.brooks@givaudan.com (J.B.); heather.carolan@givaudan.com (H.C.)
4 Polyclinique Courlancy, 51100 Reims, France; jean.tigue@me.com
5 Givaudan France Naturals, 84911 Avignon, France; benedicte.sennelier-portet@givaudan.com
* Correspondence: marie.meunier@givaudan.com

**Abstract:** Microbiome supplementation initially targeted the gut microbiota but has since been extended to the skin. A new category, psychobiotics, defined beneficial compounds with a positive action on microbiota, providing benefits to the host's mental health. *Pogostemon cablin* leaf extract, proven to alleviate scalp dryness, was clinically evaluated on volunteers presenting dry scalp with flakes. A metagenomics study and sebum production analysis were performed, coupled to flakes scoring. The benefits of *Pogostemon cablin* leaf extract on emotions were assessed through three neuroscientific methods. Through this study, we proved that the skin microbiota of dry scalp was imbalanced, with increased alpha diversity and decreased *Cutibacterium* relative abundance compared to oilier skin. After applying our ingredient for one month, microbiota was rebalanced with a decrease in alpha diversity and increase in *Cutibacterium* relative abundance compared to the initial profile. Microbiota rebalancing led to an increase in scalp sebum and decrease in dry flakes compared to the start of the study. This global rebalancing improved the emotional state of people with scalp dryness who expressed more positive emotions after treatment.

**Keywords:** scalp disorders; emotions; neurocosmetics; psychobiotics; skin-brain axis; *Pogostemon cablin*





## 1. Introduction

Product ranges for microbiome supplementation have existed for several decades and were initially designed to target the gut microbiota. Three categories of microbiome supplementation have been created and are detailed as follows: prebiotics, which are food components that positively affect the gut microbiota (e.g., lactulose, fructo-oligosaccharides and inulin); postbiotics, which are substances resulting from microbiota metabolism and that have a beneficial effect on the microbiota itself as well as the host; finally, probiotics, which are living microorganisms having health benefits for consumers [1]. These definitions, which initially described ingested products for microbiome supplementation, were recently transposed to topically applied products having a beneficial effect on the skin microbiota.

A newly formed category of products dedicated to the microbiome are psychobiotics. They are defined as beneficial bacteria or compounds that have a positive action

on beneficial bacteria, providing benefits to the host's mental health [2]. Indeed, the communication between the gut and the brain has been studied in depth, and three major communication pathways have been identified: the hypothalamic-pituitary-adrenal axis, the immune response and neurotransmitters; three pathways that also interestingly link the skin microbiota to the brain [3].

The identification of short-chain fatty acids (SCFAs) that are produced by the gut and the skin microbiome and are involved in the dialogue between microbiome and pain open new doors for the development of topically applied psychobiotics. SCFAs are saturated aliphatic organic acids that consist of one to six carbons: acetate, propionate, and butyrate are the most abundant. It has been proven for the gut microbiome that SCFAs act as metabolic substrates regulating the host cellular metabolism and differentiation, the integrity of the epithelial barrier through the synthesis of mucus, the immune system and microglia maturation. Very interestingly, microglia regulates emotional alteration through mediating immune response (modified microglia metabolism in some mental disorders, e.g., depression and bi-polar disorders) [4–6]. Recently, some research works showed that providing oligosaccharides supplementation is an efficient way to stimulate the production of SCFAs by the microbiota [7–9].

We can hypothesize a similar role on the skin for the SCFAs produced by the skin microbiota. Indeed, it has been already proven that *Cutibacterium* acnes and *Staphylococcus epidermidis* were able to produce acetate, propionate and butyrate [10]. The production of fatty acids by the skin microbiota to bring skin benefits, such as rebalancing skin sebum production, would be of great interest regarding the condition of dry scalp.

Before exploring the benefits of skin microbiota rebalancing on the condition of dry scalp, we must understand the characteristics and issues of scalp dryness. Scalp dryness is characterized by a poor sebum production by sebocytes that leads to skin barrier disruption [11]. As previously described, skin sebum level and microbiota composition are closely linked. Previous metagenomics studies performed by Mukherjee et al. [12] and by Shibagaki et al. [13] highlighted a correlation between age, sebum level of the skin and alpha diversity (Shannon index). Indeed, they demonstrated that an increase in sebum is significantly correlated with a reduction in the alpha diversity of the microbiota and with an increase in the relative abundance of *Cutibacterium* [12]. In the same way, they also showed that an increase in alpha diversity and a decrease in the relative abundance of *Cutibacterium* on skin in a group of people aged over than 60 years old compared to people aged below 37 years old [13] is consistent with the fact that skin sebum content is described to decrease during chronological ageing of the body [14]. Following these observations, we can suppose that a metagenomics analysis on dry scalp would evidence a drastic increase in alpha diversity with a weak relative abundance of *Cutibacterium*. Alpha diversity and relative abundance of *Cutibacterium* could be markers that define the scalp condition, especially in this context of scalp dryness, and could be an interesting way to highlight the biological benefits of active ingredients on dry scalp and microbiota.

In addition to poor sebum production by sebocytes, keratinocytes can present an anarchic migration through the over-stimulation of the p38/mitogen-activated protein kinase (MAPK) pathway, leading to an abnormal stratum corneum thickening [15–17].

Scalp dryness is also due to a very low level of hydration, causing the production of dry flakes [18]. Skin hydration is possible due to a good balance between the maintenance of skin barrier function and the production of natural moisturizing factors (NMFs) [19]. In the context of dry skin, there is a loss of expression of the FLG gene coding for profilaggrin and filaggrin, resulting in the impairment of skin barrier function, and of hydration as a result [20]. This combination of characteristics causes scalp itching and scratching, leading to the production of dry flakes which fall in the hair and on clothes.

Beyond the biological characteristics of scalp dryness and the link with skin microbiota, falling flakes in an unsightly problem that leads to emotional distress as many unsightly issues. To understand why, experts in skin research have recently increased their interest in the skin-brain axis in order to understand the link between people's skin condition and

their state of mind. Psycho-dermatology studies have demonstrated that people suffering from skin sensitivity and, more globally, anaesthetic disorders, are emotionally affected by their skin appearance and are more prone to depression, anxiety, feel unable to play a useful role, lose confidence and experience many other negative feelings [21–24].

As described earlier, the microbiome is another contributor to overall health and wellbeing. That is why, through this study, the microbiome-skin-brain axis was explored by studying the impact of the modulation of skin microbiota composition and skin structure improvement on the feelings and the mood of people suffering from dry scalp. Indeed, we hypothesized that improving the skin sebum production and microbiota composition would improve the skin condition of people suffering from scalp dryness, until the improvement of their mood.

To help us in this research, we worked with *Pogostemon cablin* Benth. (id: wfo-0000279652), a species of plant from the family Lamiaceae that is native to the Philippines where it grows wild as in most parts of South Asian countries. The aerial parts (leaves and stems) were previously distillated to obtain the essential oil. Then, they were dried and reused through an eco-design process with Subcritical Water Extraction (SWE) to obtain an extract rich in oligosaccharides. In this work, the benefits of this extract were first evaluated in vitro and in vivo on the condition of dry scalp regarding skin structure, sebum and the production of dry flakes. In a second step, we went further by studying if these benefits could lead to microbiota rebalancing and an improvement in volunteers' feelings in order to complete the knowledge about the microbiome-skin-brain axis.

## 2. Materials and Methods

### 2.1. Pogostemon cablin Extract Preparation

#### 2.1.1. Source of Patchouli (*Pogostemon cablin* Benth.)

The plant material (including leaves and stems), collected in Sulawesi (Indonesia), was supplied by PT Indowangi Nusajaya and has been authorized by TRU-ID (Certificate of Authentication Number P001.1-NTX03212019). The harvest period is all year long. The material was dried for three days then bundled and kept for a few days before distillation. This storage allowed for a light fermentation process to release some key odor components of the plant.

#### 2.1.2. Obtention of *Pogostemon cablin* SWE Extract

Obtention of *Pogostemon cablin* Oil and *Pogostemon cablin* Exhausted Aerials Parts

The *Pogostemon cablin* aerial parts were subjected to a first steam distillation locally by the farmers and to a subsequent redistillation in stainless steel vessels to obtain *Pogostemon cablin* oil (1 kg of oil is made from 50 kg of leaves), leaving behind the exhausted *Pogostemon cablin* aerial parts. One day after distillation, the exhausted leaves and stems of *Pogostemon cablin* were dried under direct sunlight for a minimum of three days. The ratio of leaves and stems was about 70:30 (% by mass).

Obtention of *Pogostemon cablin* SWE Extract

Between 0.75 kg and 1.1 kg of dried exhausted *Pogostemon cablin* aerial parts were placed into a 5 L extraction chamber. Demineralized water was pumped through the extraction chamber by means of a high-pressure pump (HPLC) at 150 mL/min until a pressure of 20 bar was reached. The water circulating through the extraction chamber was previously heated to a temperature of 125 °C by means of a heating resistor. The temperature was measured by means of a thermocouple (K-type) connected to a temperature control. At the exit of the extraction chamber, the extract was cooled using an ice bath and then collected in a vessel at atmospheric pressure. The extraction was performed at a ratio of plant to solvent of 1:5 (ratio by mass), such that 12.75 kg of aqueous extract were obtained from 2.55 kg of plant, for instance. Two (2) kg of the thus obtained crude aqueous extract was diluted by the addition of 2 kg of demineralized water and filtered through a 1 μm filter cloth, followed by sterile filtration over a 0.2 μm filter plate to obtain 4 kg of an extract. The

dry matter was from 1 to 3%. For storage and to avoid microorganisms' contamination, 1% of phenethylalcohol was added to the final SWE extract.

### 2.1.3. Phytochemical Characterization of *Pogostemon cablin* SWE Extract

Fractionation of SWE Extract by CPC-NMR (Centrifugal Partition Chromatography—Nuclear Magnetic Resonance)

Following the method described by Hubert et al. [25], 1.8942 g of SWE dry extract was injected with a CPC instrument FCPE300® (Rousselet Robatel Kromaton) with a two-phase solvent system (Ethyl acetate, acetonitrile and water, 3:3:4 ratio by volume). The stationary phase was the lower phase of the two-phase solvent system and the mobile phase was the upper phase of the two-phase solvent system. After column equilibration at 20 mL/min, the crude extract was dissolved in 20 mL of the lower phase + 5 mL of the upper phase, and injected through a 30 mL sample loop. The mobile phase was pumped in the ascending mode for 70 min, and then the column was extruded by switching the mode selection valve for 10 min. Fractions of 20 mL were collected over the whole experiment and combined according to their thin layer chromatography profiles. TLC was performed on pre-coated silica gel 60 F254 Merck plates with the migration solvent system ethyl acetate/toluene/acetic acid/formic acid (70/30/11/11, *v/v*), visualized under UV light at 254 nm and 366 nm and revealed by spraying the dried plates with the NP reagent, and then by 50% $H_2SO_4$ and vanillin followed by heating. Twelve (12) fractions were obtained and an aliquot of each fraction from F2 to F12 (up to 20 mg when possible, the mass of F1 < 1 mg was not sufficient to perform NMR) was dissolved in 700 µL of DMSO-d6 and analyzed by $^{13}$C NMR at 298 K on a Bruker Avance AVIII-600 spectrometer (Karlsruhe, Germany) equipped with a TXI cryoprobe. After spectra processing, the absolute intensities of all $^{13}$C NMR signals were automatically collected and binned across the spectra of the fraction series by using a locally developed computer script. The resulting table was imported into the PermutMatrix software (version 1.9.3, LIRMM, Montpellier, France) for Hierarchical Clustering Analysis (HCA). The resulting $^{13}$C NMR chemical shift clusters were visualized as dendrograms on a two-dimensional map.

For metabolite identification, each $^{13}$C NMR chemical shift cluster obtained from HCA was manually submitted to the structure search engine of the database management software (ACD/NMR Workbook Suite 2012, ACD/Labs, Ontario, Canada) comprising the structures and predicted chemical shifts of low molecular weight natural products. In parallel, a literature survey was performed to obtain the names and chemical structures of the maximum number of metabolites already reported in the species *Pogostemon cablin*. Additional 2D NMR experiments (HSQC, HMBC, and COSY) were performed on fractions containing putatively identified compounds to confirm the molecular structures proposed by the database at the end of the dereplication process.

Quantification of Total Sugars by HPAE-PAD (High Performance Anion Exchange Chromatography with Pulsed Amperometric Detection)

About 100 mg of samples were weighted into 20 mL single tubes and 2.5 mL of water and 0.5 mL of hydrochloric acid was added. The mix was heated at 100 °C for 6 h. After cooling at room temperature and transferring the solution to a 10 mL volumetric flask, the volume was filled with water. One (1) mL of the solution was diluted in a 20 mL volumetric flask with water. The solution was passed through a 0.45 µm RC filer then through an OnGuard Ag/H cartridge before injection in the Ionic chromatography system. The column used was a Dionex Ionpack PA1 (10 µm 2×250mm) at 30 °C with a flow of 0.25 mL/min and an injection volume of 2.5 µL. The detector was a pulsed amperometer and the eluent was deionized water (A) and NaOH 200 mM in water (B). The gradient used is described in the Table 1 below:

**Table 1.** Mobile phase gradient used in HPAE-PAD for the quantification of sugars in SWE *Pogostemon cablin* extract.

| Time (min) | %A | %B |
|:---:|:---:|:---:|
| 0 | 95 | 5 |
| 20 | 95 | 5 |
| 30 | 75 | 25 |
| 40 | 0 | 100 |
| 70 | 95 | 5 |

A calibration curve with simple sugars (glucose, fructose, sucrose, rhamnose, galactose, arabinose, trehalose) was performed. Stock solution was performed with 20 mg of each standard in a 100 mL volumetric flask brought to volume with water. Then four dilutions were performed for calibration points.

Evaluation of the Molecular Weight (MW) of Polymers by Size Exclusion Chromatography (SEC)

Approximately 50–60 mg of sample was weighted and diluted with 4 mL of water. Two SEC columns were used in serial (first column 1: Polysep GFC-P4000 300 × 7.8 mm and column 2: Polysep GFC-P6000 300 × 7.8 mm from Phenomenex). The eluent used was water with a flow of 0.8 mL/min and an injection volume of 10 μL. Two detectors were used: RID (35 °C) (Refractive Index Detection) an UV at two wavelengths: 210 nm and 280 nm. A calibration curve was used with Dextran 5000, 25,000, 150,000, 410,000, 670,000. For standard preparation: 10 mg of analytical standard was weighted and diluted with 4 mL of water. CIRRUS GPC software version 3.4.1 from AGILENT (Agilent Technologies Inc., Santa Clara, CA, USA) was used to assign and determine the polymer distribution.

### 2.2. In Vitro/Ex Vivo Biological Evaluation

### 2.2.1. Cells and Skin Source

All experiments were performed on primary cells and skin explants obtained from skin surgical residues following plastic surgery. Skin explants were obtained from donors who have sustained abdominoplasty and lifting (Polyclinique Courlancy, Reims) after reading, understanding and signing an "information and no objection" form for use, for dermocosmetic research purposes, of tissues, cells, and products of the human body collected during surgery (surgical residues), aligned with articles L. 1211-2 alinéa 2, and L. 1245-2, Code de la santé publique.

### 2.2.2. Transcriptomic Analysis on Keratinocytes

Normal human epidermal keratinocytes (NHEKs), freshly isolated from a 46-year-old donor, were seeded at 300,000 cells per well in 6-well plates pre-coated with type I collagen in the presence of EpiLife medium supplemented with HKGS (Human Keratinocytes Growth Supplements) factors (Gibco®, Life Technologies, Carlsbad, CA, USA). After 48 h of culture, the cells were rinsed two times with Phosphate-Buffered Saline solution (PBS, Gibco®) and allowed to rest in basal medium overnight before being stimulated with *Pogostemon cablin* extract at 0.5% ($v/v$) in EpiLife basal medium. After 18 h of treatment, RNAs were extracted according to the Trizol method (Thermo Fisher Scientific, Burlington, MA, USA). RNA quality was controlled and a reverse transcription was performed to obtain cDNA using the Verso cDNA kit (Thermo Fisher Scientific). RT-qPCR was performed on specific pre-coated plates (Applied Biosystems, Foster City, CA, USA) designed to study transcriptomic expression of different genes involved in epidermal function with 10 ng of cDNA per well using CFX96 Touch (Biorad, Hercules, CA, USA) and Universal Taqman mix (Quantabio, Beverly, MA, USA). The relative quantification (RQ) of gene expression was

calculated according to ABL1 (ABL Proto-Oncogene 1, Non-Receptor Tyrosine Kinase) and EIF2B1 (Eukaryotic Translation Initiation Factor 2B Subunit Alpha) housekeeping genes.

### 2.2.3. Keratinocytes Migration

NHEKs were seeded in 12-well plates pre-coated with type I collagen at 200,000 cells/well in EpiLife medium supplemented with growth factors. After 48 h of incubation, cells were starved overnight with basal medium alone for untreated condition or containing HB-EGF (Sigma-Aldrich, Missouri, USA) at 10 ng/mL as a positive control for migration or *Pogostemon cablin* extract at 0.5% ($v/v$). Next morning, a vertical scratch was made in each well with a 200 μL pipette-tip, cells were rinsed 2 times with PBS (Gibco®) in order to eliminate scratched cells, and treatments were applied again. Pictures were taken in bright-field mode just after the scratch and 8 h later with an inverted microscope (Axio Observer, Zeiss, Germany). The scratch area was measured at each time with ImageJ® software version 1.44.

### 2.2.4. Stratum Corneum Thickening

Skin explants from a 70-year-old donor who underwent scalp surgery were kept in culture for 10 days. Explants were topically treated daily with *Pogostemon cablin* extract at 0.5% ($v/v$) diluted in Carbopol versus Carbopol control versus untreated control. After 10 days of culture and treatment, skin explants were fixed in formalin and hematoxylin-eosin staining was carried out in order to analyze the *stratum corneum* thickness. Thickness was measured with ImageJ® software version 1.44.

### 2.2.5. Reconstructed Human Epidermis (RHE) in Dry Condition

Reconstructed Human Epidermis (RHE) were produced for 10 days in an incubator at 37 °C and 5% $CO_2$; the culture medium was changed every other day. To induce stress due to dry condition, RHE were produced at 20% relative humidity from day 8 to day 10, while negative control RHE were produced at 100% relative humidity until day 10. After 10 days, at the end of production, RHE were treated as follows:

- negative controls were kept in culture at 100% relative humidity for 48 h, with a topical application of PBS
- positive controls were kept in culture at 20% relative humidity for 48 h, with a topical application of PBS
- *Pogostemon cablin* extract at 0.5% was topically applied on RHE kept in culture at 20% relative humidity for 48 h.

RHE morphology was evaluated by hematoxylin eosin staining. The expression of filaggrin and caspase 14 was analyzed by immunofluorescence. Natural moisturizing factors (NMF) were screened through Urocanic Acid (trans-UCA), Pyrrolidone Carboxylic Acid (PCA) and Serine dosage by an LC/MS system.

### *2.3. Clinical Evaluation*

All the subjects participating in the study gave their informed consent signed at the beginning of the study. The study was conducted according to the guidelines of the Declaration of Helsinki. This study performed on cosmetic products was within the definition of article L. 5131–1v of the French Public Health Code and is in accordance with decree n° 2017-884 of 9 May 2017, modifying some regulatory requirements concerning research involving human subjects.

### 2.3.1. Sebum Production and Scalp Microbiota Analysis in Rinse-Off Application
INCI Formula

AQUA/WATER, SODIUM LAURETH SULFATE, COCAMIDOPROPYLBETAINE, ± POGOSTEMON CABLIN LEAF/STEM EXTRACT, SODIUM CHLORIDE, PHENOXY-ETHANOL, FRAGRANCE.

Each shampoo formula had a pH comprised between 5.5 and 6.0.

Panels Description

A placebo-controlled, single blind clinical study was carried out on 40 volunteers with dry and itchy scalp. The subjects were randomly and equally divided in two groups: one of which applied a placebo shampoo (14 women and 6 men; average age $51 \pm 11$ years old) while the other applied a shampoo containing *Pogostemon cablin* extract at 0.5% (10 women and 10 men; average age $50 \pm 13$ years old). Each group of volunteers applied shampoo every other day for 28 days.

The baseline of skin sebum and metagenomics composition of volunteers presenting a dry scalp was compared with another group of 30 healthy Caucasian women (mean age $\pm$ standard deviation: $30.7 \pm 5.8$ years) presenting with oily skin (sebum > 140 $\mu$g/cm$^2$ at the forehead). This second group was evaluated only at day 0 for the baseline comparison between oily skin and dry skin.

Sebum Analysis by Sebufix® Scoring

Scalp sebum was measured at day 0 and after 28 days of application to assess whether the product under study has sebum regulating properties. In this case, the Sebufix® (CKelectronic, Cologne, Germany) was applied to the upper part of the scalp to achieve the absorption of the sebum present on the scalp.

The microcamera is a professional diagnostic piece of equipment consisting of a probe for exploring the scalp and the software for capturing and displaying images. The microcamera's function is to allow a direct, magnified view of an area of the scalp to observe its state: whether flakiness, greasiness or redness are present, etc.

The attributes under study—dandruff, redness, greasiness, etc.—were evaluated on a scale of five values. The lowest value (0) corresponds to the level "absence" of the attribute and the highest value (5) corresponds to the level "very high". The intermediate values 1, 2, 3 and 4 correspond to the levels "low", "average", "considerable" and "high", respectively.

Metagenomic Analysis

- Sampling and Preparation

Microbiota samples were collected from the volunteers using a non-invasive swabbing method, using sterile swabs moistened with a sterile solution of 0.15 M NaCl. Swabs were transferred at $-20\,^{\circ}$C and kept frozen until DNA extraction.

Sampling was performed before treatment (D0) and after 28 (D28) days of treatment, using a standardized procedure. DNA extraction was performed using the DNeasy PowerLyzer® PowerSoil® DNA Isolation Kit with a Qiacube device (Qiagen, Hilden, Germany), with the following modifications. The tip of each swab was detached with a sterile surgical blade and transferred into a 1.5 mL tube containing 750 $\mu$L of bead solution. The sampled biomass was suspended by stirring and pipetting and then transferred to a bead beating tube. The remaining steps were performed according to the manufacturer's instructions. According to the manufacturer's instructions, DNA concentration was determined using the Qubit™ dsD-NA HS fluorometric quantitation kit (Invitrogen, ThermoFisher Scientific, Courtaboeuf, France).

- Sequencing and Data Analysis

16S rRNA Gene Sequencing

Sequencing was performed with the MiSeq device (Illumina, Inc., San Diego, CA, USA) through a 500-cycle paired-end run, targeting the V3V4 16S variable regions using the following primers: 16S-Mi341F forward primer 5′-CCTACGGGNGGCWGCAG-3′ and 16S-Mi805R reverse primer 5′-GACTACHVGGGTATCTAATCC-3′, producing about 460 bp amplicons.

PCR1s were performed as follows: 8 $\mu$L of template DNA (0.2 ng) were mixed with 5 $\mu$L of each reverse and forward primers (1 $\mu$M), 5 $\mu$L of KAPA HiFi Fidelity Buffer (5X), 0.8 $\mu$L of KAPA dNTP Mix (10 mM each), 0.7 $\mu$L of RT-PCR grade water (Ambion), and

0.6 µL of KAPA HiFi hot-start Taq (1 U/µL), for a total volume of 25 µL. Each amplification was duplicated, and duplicates were pooled after amplification. PCR1 cycles consisted of 95 °C for 3 min and then 32 cycles of 95 °C for 30 s, 59 °C for 30 s, and 72 °C for 30 s, followed by a final extension at 72 °C for 3 min, with a BioRad CFX1000 thermocycler. Negative and positive controls were included in all steps to check for contamination. All duplicate pools were controlled by gel electrophoresis, and amplicons were quantified using fluorometry.

Libraries ready for analysis were then produced following the Illumina guidelines for 16S metagenomics libraries preparation. Briefly, the PCR1 amplicons were purified and controlled using an Agilent 2100 Bioanalyzer (Agilent Technologies, Santa Clara, CA, USA). To enable the simultaneous analysis of multiple samples (multiplexing), Nextera® XT indexes (Illumina) were added during PCR2 using between 15 and 30 ng of PCR1 amplicons. PCR2 cycles consisted of 94 °C for 1 min and then 12 cycles of 94 °C for 60 s, 65 °C for 60 s, and 72 °C for 60 s, followed by a final extension at 72 °C for 10 min. Indexed libraries were purified, quantified and controlled using an Agilent 2100 Bioanalyzer. Validated indexed libraries were pooled in order to obtain an equimolar mixture.

The run (500 cycles) was achieved on MiSeq sequencer (Illumina) using the MiSeq Reagent Kit v3 600 cycles (Illumina). The sequencing run produced an output of 12.7 million paired-end reads of 250 bases, i.e., up to 3.2 gigabases. The libraries and the MiSeq run were performed by Givaudan, at the GeT-PlaGe platform (INRA, Auzeville, France).

After the MiSeq run, raw data sequences were demultiplexed and quality-checked to remove all the reads with ambiguous bases. Indexes and primer sequences were removed with cutadapt (v1.9; http://cutadapt.readthedocs.io/en/stable/index.html, accessed on 15 January 2020) and reads with fastq scores lower than 28 were trimmed. The forward and reverse sequences were paired using bbmerge (https://jgi.doe.gov/data-and-tools/bbtools/, accessed on 15 January 2020). Samples with less than 5000 paired-sequences were discarded. The remaining paired sequences were then treated using an in-house pipeline that uses vsearch to remove chimeras and amplicons with PCR errors. Sequences were then split into Operational Taxonomic Units (OTUs, a cluster of similar sequence variants of the 16S rRNA marker gene sequence) at a 1% dissimilarity level using swarm (v2.6). Unique amplicons were mapped to the SILVA SSU Ref NR 99 (non-redundant) database (release 132; https://www.arb-silva.de/, accessed on 16 January 2020) for taxonomic assignation using the RDP classifier. Data normalization and analyses were performed using R statistical computing environment (v3.2.0; https://www.r-project.org/—R core team (2014) (accessed on 16 January 2020) using the Bioconductor package (mainly Phyloseq, DESeq2 and Vegan libraries; http://www.bioconductor.org/, accessed on 17 January 2020). Data were then compared using Wilcoxon's test for paired samples. Due to the use of multiple tests, *p*-values were adjusted using false discovery rate (FDR) correction.

2.3.2. White Flakes Production in Rinse-Off Application
INCI Formula

See INCI Formula Section 2.3.1.

Panel Description

A placebo-controlled, single blind clinical study was carried out on 40 volunteers aged more than 18 years old and presenting with dry and itchy scalp with the presence of white flakes. The subjects were randomly and equally divided in two groups: one of which applied a placebo shampoo (four men and sixteen women; average age 33 ± 1 years old) while the other applied a shampoo containing *Pogostemon cablin* extract at 0.5% (five men and sixteen women; average age 33 ± 2 years old). Each group of volunteers applied shampoo every other day for 28 days.

White Flakes Scoring

The clinical evaluation was performed on the head divided into four parts according to the Figure 1. Indeed, after two and four weeks of shampoo application, a technician evaluated the presence of non-adherent dandruff in each of the four parts on the top of the head. A score was attributed to each part according to the description of score meaning in the table below. The final score is the mean of the four-part scores.

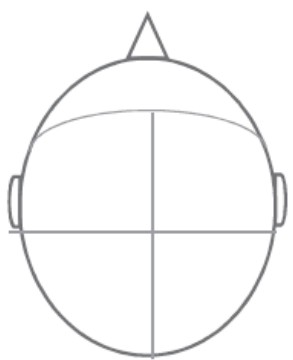

| Score | Non-adherent dandruff |
|---|---|
| 0 | No dandruff |
| 1 | A few dispersed dandruff flakes |
| 2 | A small quantity of dandruff |
| 3 | A moderate quantity of dandruff |
| 4 | A large quantity of dandruff |
| 5 | A very large quantity of dandruff |

**Figure 1.** Head scoring area was divided into four parts and the score is the mean of the four-part scores. After two and four weeks of shampoo application, a score from 0 to 5 was attributed to each part, meaning no dandruff to a very large quantity of dandruff, as described in the table.

Illustrative Pictures by C-Cube®

The C-Cube 2® (Pixience, Toulouse, France) dermoscope allows skin image acquisition in Ultra High Definition (4K UHD video streaming; 18 million pixels image resolution). It incorporates True Color patented technology, which optimally renders the full spectrum of the skin's natural colors. The size of image acquisition is $12 \times 16$ mm and the magnification is $\times 5$.

2.3.3. White Flakes Production and Emotions Improvement in Leave-On Application

INCI Formula

AQUA/WATER, ± POGOSTEMON CABLIN LEAF/STEM EXTRACT, SODIUM BENZOATE, PPG-26 BUTETH-26, PEG-40 HYDROGENATED CASTOR OIL, CITRIC ACID, FRAGRANCE, CITRONELLOL, BUTYLPHENYL METHYLPROPIONAL, D-LIMONENE, ALPHA-ISOMETHYL IONONE.

Panel Description

The study was performed in a double-blind, randomized and placebo-controlled environment. The assessment was based on an intra-subject comparison. Thirty (30) volunteers (two groups of 15 volunteers) were involved in the study, aged between 18 and 75 (average age: $43.4 \pm 13.3$ years), presenting white flakes, itching on the scalp and with a dry scalp (corneometer® < 50 a.u on the forehead/scalp) corresponding to the inclusion and exclusion criteria previously defined in the protocol. Volunteers applied products twice a day for 28 days as follows: three lines were performed on scalp where four doses of products were applied followed by massage until entire product penetration.

White Flakes Scoring

The evaluation of the quantity of white flakes was performed by scoring using the following scale from 0 to 5 as shown in the Figure 2:

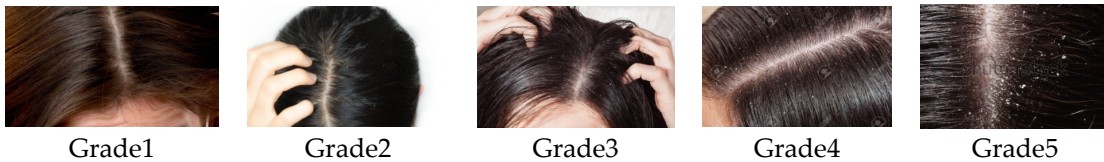

Grade1    Grade2    Grade3    Grade4    Grade5

**Figure 2.** Pictures illustrating the five grade categories used to score the severity of white flakes presence on the scalps of volunteers.

A pre-selection was carried out in order to select the volunteers presenting a score equal or greater than 1 to participate in the study. Scoring was conducted at D0 and D28 to observe the evolution. The scoring was performed on dry hair and the last use of shampoo was conducted two days prior.

Emotional Evaluation with Neuroscience Tools

- Prosody analysis

In the study, subjects were asked to verbalize about the state of their scalp. From the audio recording of their free verbal productions, an analysis of prosody was applied. The analysis of the prosody consists of capturing the vocal signal of the verbal expression of subjects and analyzing the physical parameters of the voice in terms of emotional expression.

The question that was put to them at D0 and after 28 days of using the hair lotions was as follows: "What do you feel today about your scalp?"

Using computer-based solutions, two main variables were extracted:

- Loudness (i.e., vocal intensity), represented by the mean amplitude measured in dB;
- Pitch (i.e., tone), represented by the coefficient of variation in the fundamental frequency (cvF0) measured in Hz.

The combination of these two variables allows for the assessment of the degree of emotional valence and arousal expressed vocally.

- Picture selection method

The leverage picture selection method was utilized to measure the emotional responses of consumers after using photos and applying a hair lotion containing the active ingredient and a placebo lotion. All participants completed the picture selection method on day 0 (baseline) and on day 28 before being assessed in the clinical trial.

As part of this study at D0, the volunteers were asked a question with pictures hanging on the wall: "Take a moment to think about the condition of your current scalp. What emotions come to mind then? Choose the photos that best represent these emotions. Select as many photos as you want and take as much time as you need".

After using the shampoo for more than 28 consecutive days, volunteers were invited to answer the same question with the pictures on the wall.

- Non-verbal communication

The measure of the unconscious emotional power of the lotion on the consumers was carried out by an expert in non-verbal communication. Inspired by US Psychologist Paul Ekman's work, demonstrating emotions drive non-verbal reactions, and in a universal way. The purpose of the study was to determine the emotional impact of the active lotion compared to a placebo lotion containing no active ingredient. The emotional impact is the ability of the lotion to generate positive and negative emotions following the application of it by a group of users.

To determine the emotional impact of the active lotion, the body language of 14 users who used placebo lotion was compared to that of 15 users who used lotion containing *Pogostemon cablin* extract.

Users were therefore interviewed and filmed face to face for about 5 min on their use of the lotion. A study of their expressiveness had been organized before the study to check their normal abilities to express their emotions with their faces.

### 2.4. Statistical Analysis

For in vitro, ex vivo and clinical data, data normality was first verified via the Gaussian law using the Shapiro–Wilk test. According to the results, parametric or nonparametric tests were used to compare the effect of *Pogostemon cablin* extract versus the untreated condition or versus placebo, with # $p < 0.1$; * $p < 0.05$; ** $p < 0.01$ and *** $p < 0.001$.

## 3. Results and Discussion

### 3.1. Evaluation of the Correlation Scalp Sebum/Scalp Microbiota

To begin this study, the correlation between sebum level and scalp microbiota composition was evaluated by performing a metagenomics analysis to compare microbiota between dry and oily scalp.

First, the link between sebum level and scalp microbiota composition was studied using the 16S sequencing approach. An interesting significant variation on scalp microbiota composition which is related to the amount of sebum was observed (Figure 3A). Among them, there was an important +54% increase of the relative abundance of *Cutibacterium* on oily scalp. The high dominance of *Cutibacterium* on oily scalp could be linked with the demonstration that *Cutibacterium* are able to produce SCFAs that increase the metabolism of skin sebocytes, thus supplying the scalp with fatty acids and maintaining the oily state of the scalp [10,26]. A high relative abundance of *Cutibacterium* should be used as marker to evaluate the impact of microbiota composition on scalp sebum.

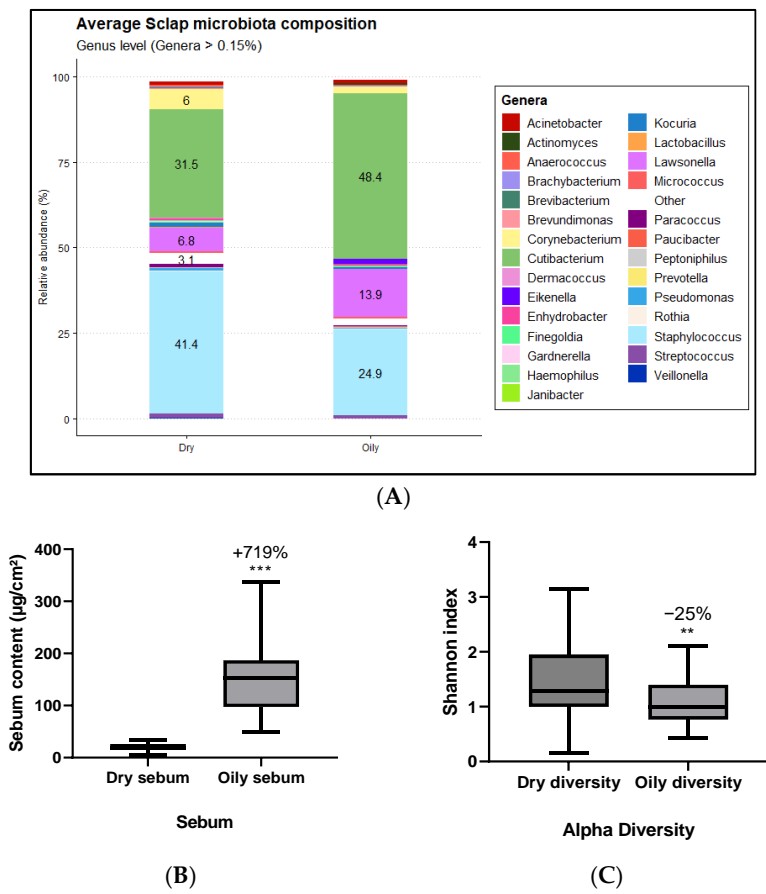

**Figure 3.** Relative abundances of bacterial genera were identified in significantly different proportions between groups with dry and oily scalp. These proportions represent a percentage of the total reads obtained by the 16S sequencing (**A**). Sebum content (**B**) and Shannon index distribution (**C**) for the two groups were defined based on skin sebum content. The significance between dry and oil scalp was calculated by applying a statistical Wilcoxon test with ** $p < 0.01$ and *** $p < 0.001$.

The link between the alpha diversity (Shannon index) and the level of sebum was also studied. The alpha diversity is based on an index giving information about the global microbiota composition. The more that alpha diversity is elevated, the more we expect to observe a complex microbiota composition, and conversely. A significant reduction of alpha diversity between dry and oily scalp by −25% was shown, correlated with a huge increase of sebum production by +719% between dry and oily scalp (Figure 3B). The scalp sebum production is known to induce *Cutibacterium* proliferation, and this proliferation will in turn activate the production of sebum [27,28].

These results confirmed that the composition of scalp microbiota has an impact on the quantity of sebum on scalp, and conversely.

Here, we demonstrated that the level of sebum, the alpha diversity and the relative abundance of *Cutibacterium* are three biomarkers that can highlight the effect of active cosmetics on dry scalp. Indeed, a dry scalp presents a low level of sebum linked to an important alpha diversity and a low relative abundance of *Cutibacterium*. Conversely, the oily scalp shows an important amount of sebum related to a low alpha diversity with a high relative abundance of *Cutibacterium*. Following these observations, an active ingredient from microbiota composition was developed to provide benefits to volunteers exhibiting dry scalp condition until they presented mood improvement.

### 3.2. Characterization of Pogostemon cablin SWE Extract Composition

In order to characterize the composition of our SWE extract, a fractionation by CPC (Centrifugal Partition Chromatography) was performed. Twelve (12) fractions were obtained and then each fraction was analyzed by $^{13}$C NMR. Through an HCA (Hierarchical Clustering Analysis), the resulting $^{13}$C NMR chemical shift clusters were visualized as dendrograms on a two-dimensional map. For metabolite identification, each result was manually submitted to the structure search engine of the database management software ACD/NMR version 12.0 comprising the structures and predicted chemical shifts of low molecular weight natural products. Polyphenolic compounds (flavonoids, phenolic acids) and organics acids were the major compounds identified in the fractions until F9, then the fractions recovered over the extrusion step (F10-F12, 85% of the extract mass) were not fully soluble in DMSO-d6 and were probably composed of oligosaccharides.

Then, the composition of sugars (free and total sugars of our SWE extract) was analyzed. The results in Table 2 demonstrated that after hydrolysis the content of sugars was multiplied by 24. These results confirm our previous hypothesis on the presence of oligo/polymers of sugars. We also demonstrated that the dry matter was composed of close to 50% of sugars.

**Table 2.** Quantification of free and total sugars in % on dry basis by HPAE-PAD in SWE *Pogostemon cablin* extract.

| Type of Extraction | Free Sugars Content in % on Dry Basis (HPAE Analyses) | Total Sugars in % on Dry Basis (HPAE Analyses) |
|---|---|---|
| SWE extract—*Pogostemon cablin* extract | 1.9 | 45.6 |

To complete, a size exclusion chromatography coupled to RID allowed for the determination of the size of the oligo/polymers. UV detection was also used in order to check the presence of chromophore that could confirm the presence of amino acids in the polymer structure. The results in Table 3 demonstrated that 28% of the oligomers were not detected in UV and were only detectable in RID, suggesting that they could be considered as oligosaccharides. The remaining 72% of oligomers were detected in UV at 210 and 280 nm, suggesting they could be identified as proteoglycans.

**Table 3.** Relative % repartition (area basis), molecular weight (MW) in Da and UV detection observed in the polymer structure identified in the SWE *Pogostemon cablin* extract.

| Relative % (Area)-Qualitative Repartition | MW in Da | UV Detection at 210 and 280 nm |
|---|---|---|
| 28% | From 569 to 1868 Da | No |
| 50% | From 22,967 to 71,058 Da | Yes |
| 22% | From 470,687 to 1,462,727 Da | Yes |

*3.3. Gene Regulation*

After developing an extract from *Pogostemon cablin* leaves rich in oligosaccharides, the potential benefit of this active ingredient was first evaluated through a transcriptomic study. Transcriptomic analysis was performed by RT-qPCR on specific pre-coated plates designed to study the transcriptomic expression of different genes involved in epidermal function, such as epidermal cohesion, epidermal differentiation, barrier homeostasis, DNA repair and skin calming. The results were expressed in comparison with those from the untreated condition used as negative control and were normalized with the average of most stable housekeeping genes (ABL1 and EIF2B1). The values of fold change and their SEM were reported in the Table 4.

**Table 4.** Gene expression was expressed as a fold change after 18 h of stimulation with *Pogostemon cablin* extract at 0.5% in comparison with the untreated control. The significance was calculated by applying a statistical Student *t*-test.

| Genes | Fold Change | SEM | *p* Value |
|---|---|---|---|
| CLDN1 | 1.489 | 0.14 | 0.0486 |
| OCLN | 1.614 | 0.03 | 0.0001 |
| SPRR1A | 2.079 | 0.13 | 0.0023 |
| KRT13 | 1.600 | 0.04 | 0.0004 |
| FLG | 2.090 | 0.09 | 0.0006 |
| CASP14 | 12.280 | 1.23 | 0.0017 |
| CD44 | −1.421 | 0.05 | 0.0078 |
| HBEGF | −1.457 | 0.03 | 0.0009 |
| AQP3 | 2.774 | 0.5 | 0.0455 |

*Pogostemon cablin* extract showed significant bioactivity at the epidermis level by significantly stimulating genes controlling epidermal cohesion (CLDN1 +49%, OCLN +61%), differentiation (SPRRIA +108%, KRT13 +60%, FLG +109%, CASP14 +1128% and CD44 −42%), hydration (AQP3 +177%) and by inhibiting genes involved in keratinocytes migration (HBEGF −46%). Indeed, keratinocytes migration capacity is linked to the expression of heparin-binding epidermal growth factor (HB-EGF) that provokes the EGF receptor phosphorylation, leading to the phosphorylation of p38/mitogen-activated protein kinase (MAPK) [15]. MAPK is an important kinase responsible for cytoskeleton reorganization that promotes cell migration [16].

It has been described that the formation of white flakes is correlated to an excessive keratinocytes migration promoting a thickening of *stratum corneum* which is correlated to dry scalp inducing itching and micro-inflammation [17]. Due to mechanical processes, the itching strongly contributes to the accumulation of white flakes and scalp discomfort [18,29]. This transcriptomic study on keratinocytes showed that *Pogostemon cablin* extract could lead to a potential improvement of skin barrier function and a decrease in dry white flakes production, by controlling keratinocytes hyper-migration by inhibiting the expression of the HBEGF gene, an improvement of epidermis cohesion through the upregulation

of CLDN1, OCLN, SPRRIA, KRT13, FLG, CASP14 an CD44 genes and increasing scalp hydration (AQP3 gene upregulation). As the formation of white flakes is correlated to an excessive keratinocytes migration promoting a thickening of *stratum corneum*, these results suggest that *Pogostemon cablin* extract could help to fix this issue.

### 3.4. Keratinocytes Migration

Following the transcriptomic study that showed the inhibition of the HBEGF gene by *Pogostemon cablin* extract, the consequence on keratinocytes migration property was then evaluated. For that, the percentage of wound closing was measured by a scratch assay performed on keratinocytes monolayer. First, a significant increase of keratinocytes migration (+132%) was observed after 8 h with HB-EGF treatment. This is consistent with the literature that describes HB-EGF to activate the EGFR signaling pathway and stimulating p38/MAPK that is responsible for cytoskeleton reorganization and cell migration [15,16]. Contrariwise, *Pogostemon cablin* extract totally inhibited keratinocytes migration by 100%, which is consistent with the observed inhibition of the HBEGF gene (Figure 4).

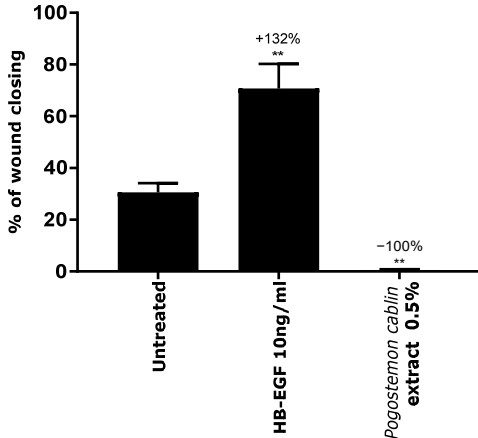

**Figure 4.** Evaluation of the inhibition property of *Pogostemon cablin* extract on keratinocytes migration. The significance of the impact of products on keratinocytes migration was calculated by applying a statistical Student t-test with ** $p < 0.01$.

This result showed that *Pogostemon cablin* extract would have a positive impact on disorders presenting important and anarchic keratinocytes migration, such as the formation of white flakes.

### 3.5. Stratum Corneum Thickness Analysis

Anarchic keratinocytes migration is associated with an important *stratum corneum* thickening which then leads to the formation of white flakes [17]. We wanted to evaluate if the keratinocytes migration inhibition by *Pogostemon cablin* extract could reduce *stratum corneum* thickening and thus lead to an improvement regarding the formation of white flakes.

After 10 days of incubation, a significant *stratum corneum* thickening of +59% was observed without any treatment compared with D0. This model mimics *stratum corneum* thickening that leads to the formation of white flakes. When Carbopol vehicle was topically applied, *stratum corneum* presented the same thickness as untreated condition, demonstrating that it had no impact on it. The topical application of *Pogostemon cablin* extract significantly decreased *stratum corneum* thickness by −35% compared to Carbopol condition (Figure 5).

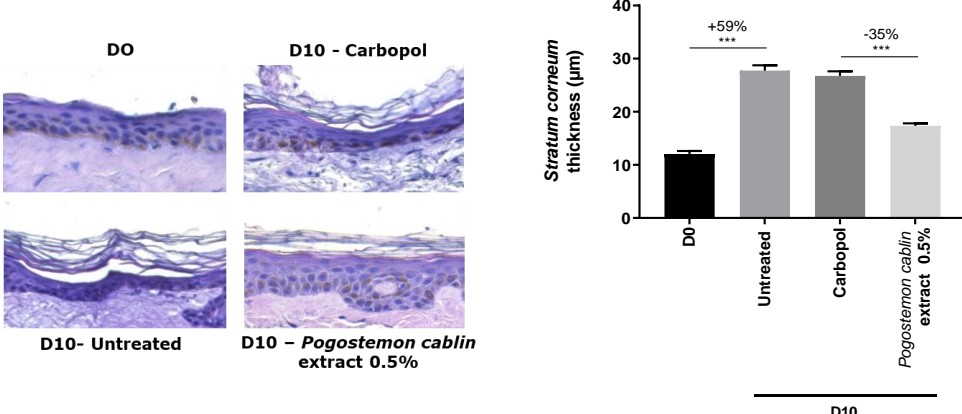

**Figure 5.** Impact of *Pogostemon cablin* extract on *stratum corneum* thickness. Thickness was naturally induced by a long-time culture of skin explant. Hematoxylin-eosin staining was carried out (left panel) and *stratum corneum* thickness was measured on pictures (right panel). The significance of the impact of products was calculated by applying a statistical Mann-Whitney test with *** $p < 0.001$.

Here, we confirmed that *Pogostemon cablin* extract, through the inhibition of HBEGF expression and keratinocytes migration, was able to restore a normal *stratum corneum* thickness.

*3.6. Reconstructed Human Epidermis in Dry Culture Condition*

The transcriptomic analysis also showed the stimulation of genes involved in epidermis differentiation (SPRRIA, KRT13, FLG, CASP14 and CD44) after treatment with *Pogostemon cablin* extract, suggesting this extract could effectively stimulate the natural tools of the epidermis to restore a strong barrier function and thus lead to an improvement of dry skin condition.

To answer this question, *Pogostemon cablin* extract was evaluated on an innovative RHE model cultivated at a low level of hygrometry for 48 h leading to RHE adaptation mimicking dry skin features. For that, RHE were exposed to low hygrometry for 48 h with or without topical application of *Pogostemon cablin* extract.

In physiological condition, skin has the capacity to maintain this balance thank to keratohyalin granules that are present in the granular layer of the epidermis. These keratohyalin granules contains the precursor pro-filaggrin that is cleaved by caspase 14, thus producing monomeric filaggrin that binds to and condense the keratin cytoskeleton to promote corneocytes compaction process for and impermeable barrier, and NMFs that contribute to the maintenance of skin hydration [30].

Here, exposure to dry condition showed a decreased number of keratohyalin granules in the epidermis that resulted in a significant decrease of filaggrin expression ($-24\%$) in the upper layers of the epidermis [30]. In normal conditions, caspase 14 is described to cleave filaggrin in order to produce natural moisturizing factors (NMF) [30]. Here, in the dry condition, there was a significant decrease of caspase 14 expression ($-24\%$) compared to the normal condition, especially in *stratum granulosum*, leading to a significant decrease of NMF production ($-29\%$) as shown in Figure 6.

The topical application of *Pogostemon cablin* extract on dry RHE increased the keratohyalin granules content in the epidermis, suggesting an increase in pro-filaggrin content. Indeed, filaggrin immunostaining showed a significant increase of filaggrin ($+56\%$) and also an increase of caspase 14 expression in dry RHE ($+49\%$) with *Pogostemon cablin* extract. This led to a cleavage of filaggrin as suggested by the significant increase of NMF content ($+35\%$) compared to the untreated dry condition.

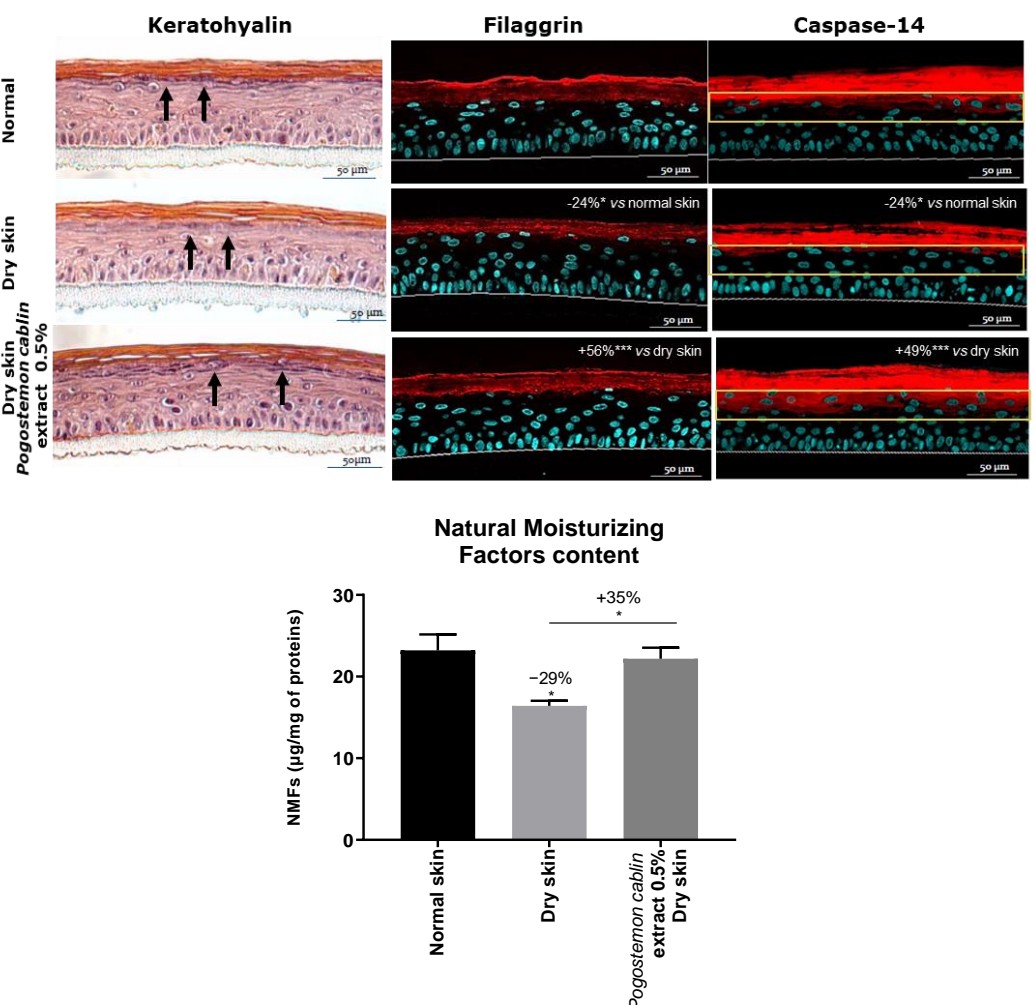

**Figure 6.** Impact of *Pogostemon cablin* extract on keratohyalin granules production (designed by black arrows), filaggrin and caspase 14 expression and natural moisturizing factors (NMF) production in dry skin condition. The significance of the impact of products was calculated by applying a one-way ANOVA followed by Dunnett test for filaggrin and caspase 14 expression with * $p < 0.05$ and *** $p < 0.001$, and a Mann-Whitney test for NMF production with * $p < 0.05$.

This model showed that *Pogostemon cablin* extract confers a protection to the skin against dryness. Moreover, these results demonstrated that *Pogostemon cablin* extract is also able to restore normal skin hydration in dried RHE condition.

### 3.7. Clinical Improvement of Microbiota Composition for a Restoration of Sebum Production

Two clinical studies were performed with a shampoo formula containing 0.5% of *Pogostemon cablin* extract versus a placebo in order to evaluate the efficacy of our active in rinse-off application on volunteers presenting dry scalp, itching, scalp discomfort and white flakes. Volunteers applied shampoo every other day for 28 days; white flakes were analyzed by scoring and were visualized by illustrative pictures obtained by C-cube® to confirm the biological benefit of the active ingredient on dry scalp. Metagenomics sampling was performed in order to study the scalp microbiota in these specific conditions. Finally, sebum was analyzed by Sebufix® where lipids droplets were evaluated by scoring performed by an expert.

### 3.7.1. White Flakes Scoring

After showing the benefits of *Pogostemon cablin* extract in vitro and ex vivo, the impact of the active ingredient was evaluated on the production of white flakes. Before any product

application and after two weeks and four weeks of shampoo application, a technician evaluated the presence of non-adherent dandruff in each of the four parts on the top of the head, as previously described in Figure 1. A final score, that is the mean of the four-parts scores, was attributed to each volunteer at day 0 and after the application of placebo shampoo or shampoo containing *Pogostemon cablin* extract.

The results showed that the application of shampoo containing *Pogostemon cablin* extract at 0.5% significantly reduced the number of white flakes with time-effect in comparison with day 0 as observed by the reduction of −23% and −33% after 14 days and 28 days of application, respectively. Moreover, the effect of *Pogostemon cablin* extract was significantly better in comparison with the placebo, showing a significant reduction of white flakes by up to 14% during the study (Figure 7A). C-Cube® was used to take pictures of the scalp area in order to better visualize the effect on white flakes. As shown in the following pictures, there was a drastic reduction in white flakes on the scalp after 14 and 28 days of active shampoo application containing *Pogostemon cablin* extract, while the placebo shampoo showed a very slight effect (Figure 7B).

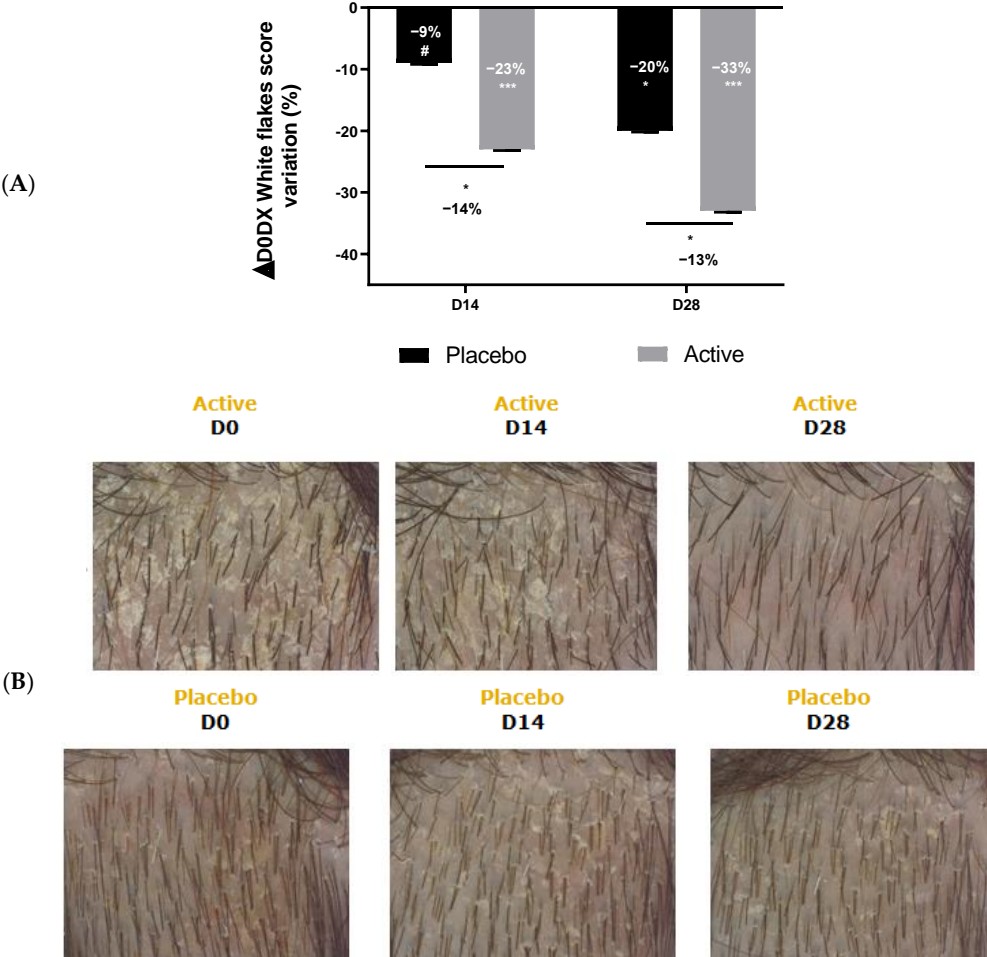

**Figure 7.** Impact of *Pogostemon cablin* extract on white flakes production on scalp from volunteers suffering from scalp dryness. (**A**) White flakes were evaluated and scored by an expert. The significance of the effect of both shampoo after 28 days of application versus day 0 was calculated by applying a Wilcoxon test with # $p < 0.1$, * $p < 0.05$ and *** $p < 0.001$. The effect of *Pogostemon cablin* extract versus placebo was calculated by applying a statistical Mann-Whitney test with * $p < 0.05$. (**B**) White flakes were illustrated with C-Cube® at day 0, after 14 and 28 days of application of a shampoo containing *Pogostemon cablin* extract (**upper panel**) or a placebo shampoo (**lower panel**).

These effects demonstrated that *Pogostemon cablin* extract was able to significantly reduce white flakes on dry scalp in rinse-off application. This efficacy must be explained by the in vitro and ex vivo results that showed keratinocytes migration inhibition and the reduction of *stratum corneum* thickness, confirming the works of Pastar et al. [17].

### 3.7.2. Metagenomics Analysis

A preliminary study showed a variation in microbiota alpha diversity between dry and oily scalp, correlated with a different relative abundance of *Cutibacterium* on these two types of scalp. Knowing that, we were interested in studying the impact of shampoo containing 0.5% of *Pogostemon cablin* extract or placebo on the alpha diversity using Shannon index after 28 days of application. The active ingredient triggered a decrease of the alpha diversity by −15% versus placebo which had a slight effect on the diversity (Figure 8A).

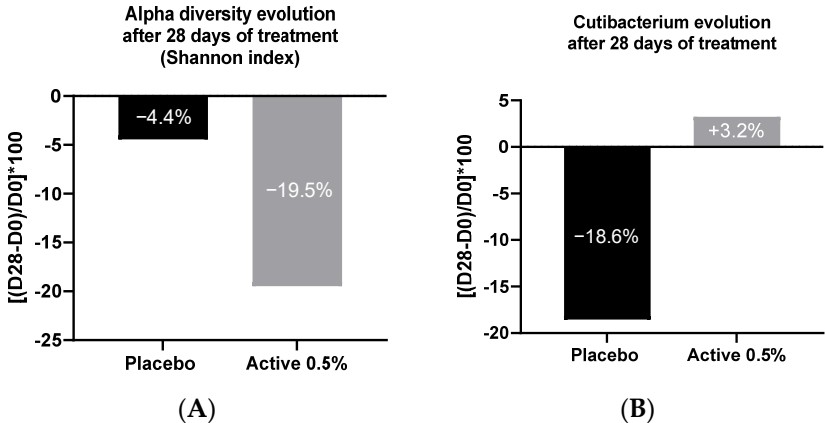

**Figure 8.** Evolution of dry scalp alpha diversity (Shannon index) (**A**) and of the relative abundance of *Cutibacterium* (**B**) after application of *Pogostemon cablin* extract at 0.5% every other day for 28 days.

To understand this variation in alpha diversity, the impact of *Pogostemon cablin* extract on *Cutibacterium* relative abundance was analyzed after 28 days of application, previously identified as a biomarker of dry scalp improvement. An important decrease of *Cutibacterium* proportion (−18.5%) was shown after 28 days of placebo application while the active ingredient induced an increase of the relative abundance of *Cutibacterium* by +22% in comparison with the placebo formula (Figure 8B). Our results confirmed that the variation in alpha diversity is correlated with the variation in the relative abundance of *Cutibacterium* on scalp, which raises questions about the amount of sebum produced by the scalp with the active ingredient.

### 3.7.3. Sebum Production on Scalp by Sebufix®

We went further by studying the link between *Cutibacterium* modulation by the active ingredient and the scalp sebum quantity. After 28 days of shampoo application, placebo shampoo decreased the sebum by −25% in comparison with D0 as shown in Figure 9, while shampoo containing 0.5% of *Pogostemon cablin* extract slightly increases it by +9%. In addition, we observed a significant improvement of sebum production with our active shampoo showing an increase of +34% in comparison with the effect of placebo shampoo.

These effects were also confirmed by illustrative pictures showing the lipid droplets caused by sebum on Sebufix®. Indeed, an increase in lipid droplets was observed after 28 days of active shampoo application while the placebo decreases them as observed by the distribution of dark spots in the illustrative pictures.

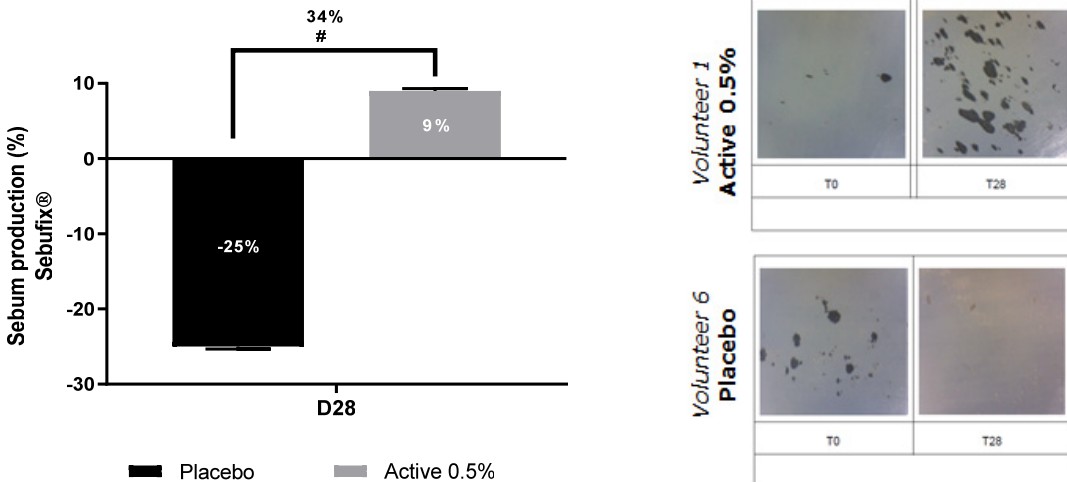

**Figure 9.** Impact of *Pogostemon cablin* extract on sebum production on scalp from volunteers suffering from scalp dryness. Lipids droplets caused by sebum were detected and illustrated with Sebufix®. The significance of the effect of shampoo containing the active versus placebo shampoo was calculated by applying a statistical Mann-Whitney test with # $p < 0.1$.

Altogether, our results confirmed a clear link between the relative abundance of *Cutibacterium* on the scalp and sebum production by the skin. Here, we showed that the application of a shampoo containing 0.5% *Pogostemon cablin* extract started sebum production by the scalp that promoted the proliferation of *Cutibacterium* as described by Mukherjee et al. [12]. As we know, *Cutibacterium* is described to be a stimulator of sebum production through the SCFAs release [28]. Thanks to the activation of sebum production by our active ingredient, it provides an optimal support for *Cutibacterium* proliferation on the scalp. We can assume that sebum production by dry scalp will be self-sustaining due to the capacity of *Cutibacterium* to produce SCFAs until the following *Pogostemon cablin* extract application. These observations suggest, for the first time, the potential existence of a virtuous circle between sebum production which enhances *Cutibacterium* proliferation and *Cutibacterium* which maintains the sebum production thanks to SCFAs release. Finally, these results also clearly indicated that the active ingredient improved dry scalp for a better scalp condition compared to normal scalp.

### 3.8. Clinical Improvement of Emotions through Microbiota Balancing

A third clinical study was performed with a lotion containing 0.5% of *Pogostemon cablin* extract versus a placebo lotion and the efficacy of the active ingredient was evaluated in leave-on application on volunteers presenting dry scalp, itching, scalp discomfort and white flakes. Volunteers applied the hair lotion twice daily for 28 days, white flakes were analyzed by scoring and volunteers' emotions were evaluated by prosody analysis, the picture selection method and non-verbal communication.

The aim of this clinical study was to observe if the improvement of scalp microbiota composition and the upregulation of sebum production could lead to an improvement of volunteers' emotions, similarly to the short-chain fatty acids that act as metabolic substrates on microglia that are able to regulate emotions [4–6].

### 3.8.1. White Flakes Scoring

After 28 days of application with hair lotion containing 0.5% *Pogostemon cablin* extract, a significant reduction of white flakes (−21%) was observed in comparison with D0. The application of the placebo formula in the same condition for 28 days showed a slight increase of white flakes (+10%) in comparison with D0. The comparison between the active ingredient and placebo lotion showed a significant reduction of white flakes (−31%) with active lotion in comparison with placebo lotion after 28 days of application (Figure 10).

These results demonstrated the strong efficacy of *Pogostemon cablin* extract even in leave-on applications on the reduction of white flakes from dry and itching scalp.

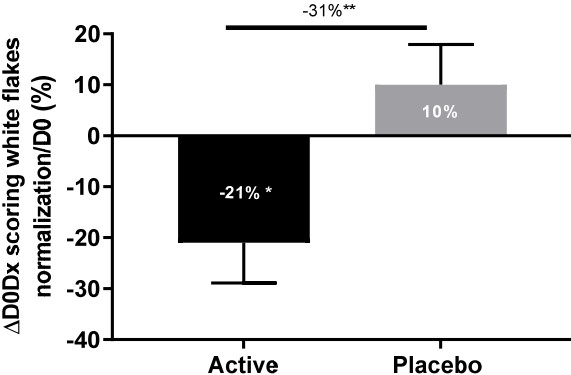

**Figure 10.** Impact of *Pogostemon cablin* extract on the production of white flakes on scalp from volunteers suffering from scalp dryness. White flakes were evaluated and scored by an expert. The significance of the effect of both lotions after 28 days of twice-daily application versus day 0 was calculated by applying a Wilcoxon test with * $p < 0.05$. The effect of *Pogostemon cablin* extract versus placebo was calculated by applying a statistical Mann-Whitney test with ** $p < 0.01$.

3.8.2. Emotions Evaluation

An emotional evaluation was performed using various neuroscience methods such as prosody, the picture selection method and non-verbal communication. Indeed, the hypothesis that people who present dry scalp with itching and excessive white flakes would be emotionally affected was presented. We supposed that having a dry and itching scalp can be related to negative mood and have an impact on self-esteem, as it has been demonstrated for people suffering from skin sensitivity [31].

Prosody Analysis

To analyze prosody, subjects were asked to verbalize about the state of their scalp at D0 and after 28 days of application of hair lotion. The variation in pitch and tonality after 28 days was compared with D0 to study the emotional response after the application of products.

After 28 days of application, active lotion increased the frequency and amplitude more than with placebo lotion. In addition, the active lotion demonstrated a significant increase of vocal frequency and amplitude in comparison with placebo, by +15.4% and + 12.4%, respectively (Figure 11).

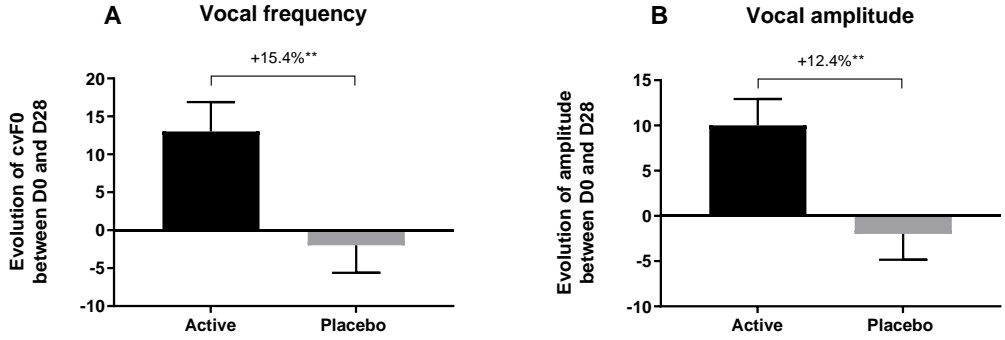

**Figure 11.** Prosody analysis before and after 28 days of application of placebo lotion or lotion containing *Pogostemon cablin* extract. Evaluation of vocal frequency (**A**) and vocal amplitude (**B**) with each product relative to day 0. The significance of the effect of both lotions after 28 days of twice-daily application versus day 0 was calculated by applying a Wilcoxon test with ** $p < 0.01$.

An increase in vocal frequency and amplitude in this context suggest that volunteers who applied active lotion felt more positive emotions than those applied placebo lotion. We can conclude that *Pogostemon cablin* extract in lotion was able to significantly improve emotional response after 28 days of application as observed by prosody evaluation.

To complete this analysis, the emotional distribution considering stimulating/unstimulating and pleasant/unpleasant emotions scale was analyzed. After 28 days of active lotion application, volunteers triggered stimulating and pleasant emotions while the placebo lotion triggered neutral emotions as shown by its location in the center of emotional valence (Figure 12A). This representation confirms that *Pogostemon cablin* triggered positive emotions contrary to placebo lotion.

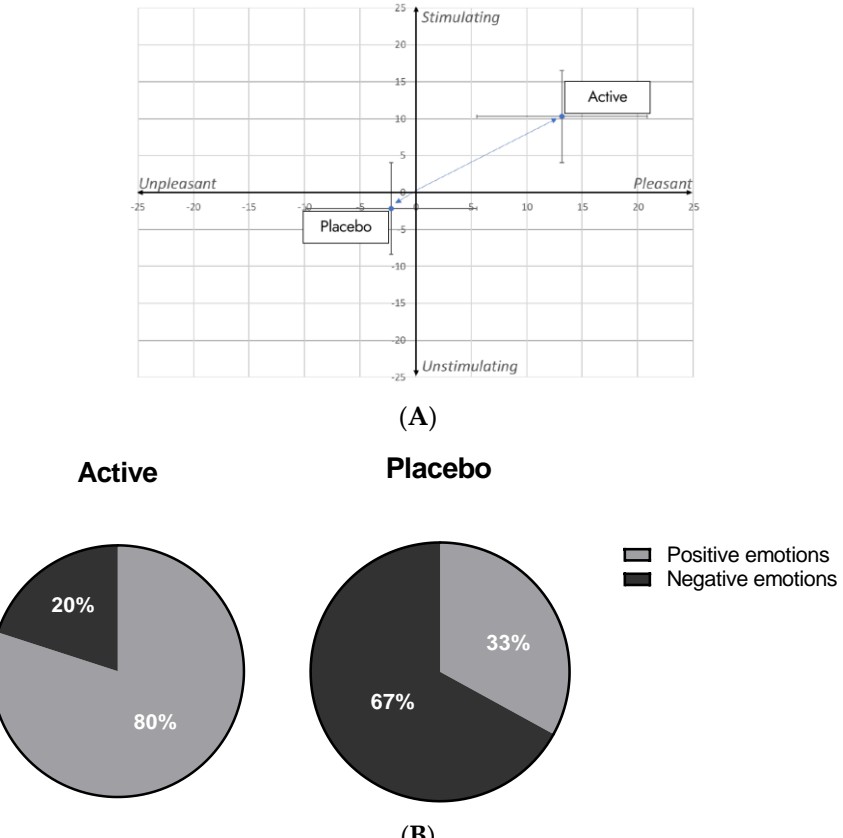

(**A**)

(**B**)

**Figure 12.** Analysis of the emotional distribution after 28 days of application of placebo lotion or lotion containing *Pogostemon cablin* extract. (**A**) Expressed emotions were evaluated according a stimulating/unstimulating and pleasant/unpleasant scale. (**B**) Number of positive and negative emotions were quantified after 28 days of application of each product.

Finally, the percentage of volunteers who have triggered negative and positive emotions for both groups was analyzed. Results demonstrated that volunteers who applied active lotion triggered 80% of positive emotions and 20% negatives. In contrast, volunteers who applied placebo lotion have triggered only 33% of positive emotions and 66% negatives (Figure 12B). Thereby, this analysis reinforces that *Pogostemon cablin* extract was able to provide positive emotions for 28 days of applications.

Non-Verbal Communication

The non-verbal communication after active and placebo lotions application for 28 days was analyzed. This method used a unique grid to analyze over 200 non-verbal reactions (facial reactions, postures, gestures, voice), and remove all verbalization barriers. After 28 days of active lotion application, 19 positives emotions and only four negatives emotions were triggered. In contrast, the volunteers who applied placebo lotion presented inverse

emotional response with 14 negatives emotions and only two positives emotions after 28 days (Figure 13). These results demonstrated that *Pogostemon cablin* extract was able to deliver a positive emotional response in comparison with the placebo lotion. Interestingly, a statistical analysis of the emotional distribution between both products and our results showed that the distribution of emotions is statistically different, demonstrating that *Pogostemon cablin* extract significantly improves emotional response in comparison with the placebo.

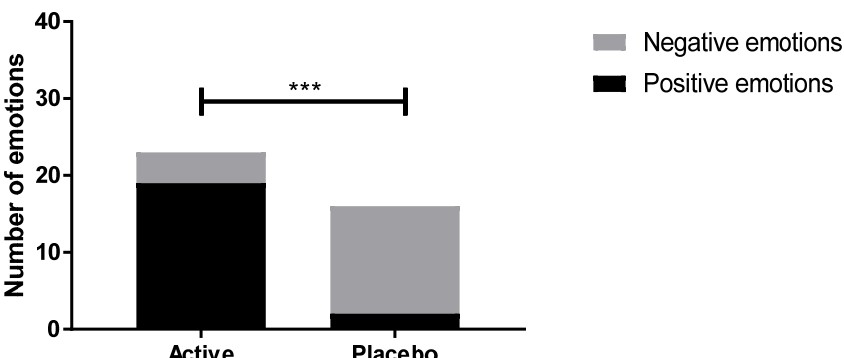

**Figure 13.** Impact of *Pogostemon cablin* extract on nonverbal communication after 28 days of application. The effect of *Pogostemon cablin* extract was compared with the placebo effect using a $\chi^2$ test with *** $p < 0.001$.

This method also allowed for identifying the type of emotions expressed after product application. Indeed, volunteers who applied active lotion delivered positive emotions including comfort, care and happiness while volunteers who applied placebo lotion expressed more negative emotions such as annoyance.

Picture Selection Method

This method showed that participants from both groups felt very negative, and significantly not happy, when thinking about their scalp before they started the clinical trial (Figure 14A). This result was consistent with the observation obtained with the two previous methods and also shows the homogeneity of emotions between the two groups.

After 28 days of lotion application, volunteers who applied the active lotion felt significantly less negative and significantly more refreshed. However, volunteers who applied the placebo lotion did not feel as negative about their scalp at day 28, but did not express positive emotions either. The results for the placebo group are neutral and reflect the placebo effect (Figure 14A).

This effect can be also visible through the list of images selected by volunteers. When volunteers thought about their scalp condition at D0, they illustrated it with images having negative and non-happy connotations (Figure 14B). After 28 days of active lotion application, there was a strong decrease in negative images associated with new images with refreshing and self-confidence connotations. The group who applied placebo lotion did not show the same efficacy as shown by the list of selected images which are mainly still negative.

This third method proved that *Pogostemon cablin* extract delivered positive emotional response after 28 days of application on volunteers who have dry itching scalp with white flakes. This mood improvement was correlated to a significant reduction in white flakes as previously demonstrated.

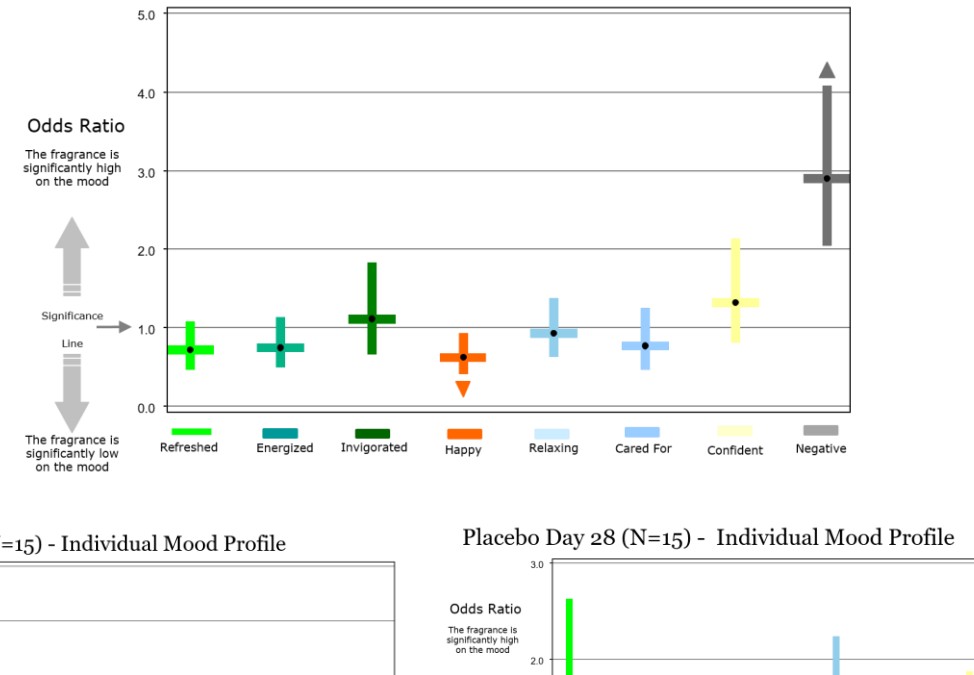

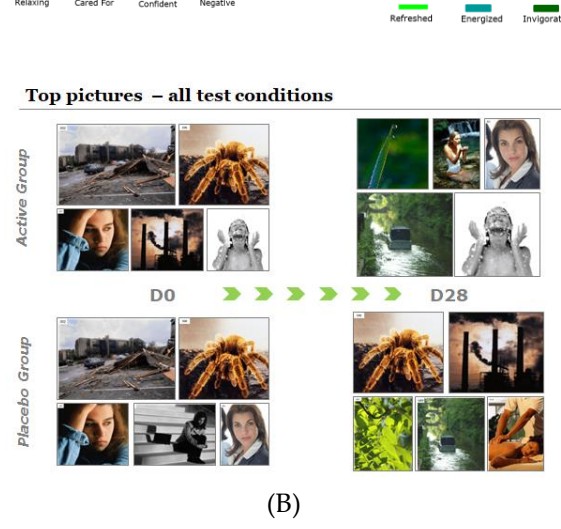

**Figure 14.** Evaluation of the impact of *Pogostemon cablin* extract on volunteers' mood using the picture selection method before and after 28 days of treatment. (**A**) Quantification of the main moods felt by volunteers. (**B**) The upper panel represents the main pictures selected by volunteers to evoke the evolution of their mood about their scalp treated with *Pogostemon cablin* extract, and the lower panel represents the main pictures selected by volunteers who applied placebo lotion.

By using three complementary methodologies based on neuroscience, we answered our hypothesis that people suffering from scalp dryness were emotionally affected. This result was consistent with psycho-dermatology studies which showed that people suffering from unpleasant experiences such as dry or sensitive skin, and with an unsightly skin disorder were more prone to depression, anxiety and many other negative feel-

ings [23,24]. Still, with these neuroscience methodologies, we showed that the application of *Pogostemon cablin* extract, which biologically improved the skin barrier and sebum production until reducing the amount of white flakes on the scalp, led to a clear improvement of volunteers' mood through the expression of positive emotions. Observing these results, two hypotheses can be presented: the first is that this mood improvement is directly linked to the biological efficacy of the product, from gene expression until a reduction of keratinocytes migration, a reduction in *stratum corneum* thickness, a restauration of barrier function and sebum production, leading to a rebalancing of scalp sebum and reduction in the production of white flakes. Following this hypothesis, volunteers feel better just by not seeing the unsightly issue of white flakes anymore. The second is that *Pogostemon cablin* extract, in addition to having biological benefits, is able to rebalance the scalp microbiota composition and sebum production, bringing it closer to a 'normal' scalp [12,13]. Indeed, the reactivation of sebum production creates an enabling environment for *Cutibacterium acnes*, but this bacterium is also known to produce SCFAs that are already described to positively affect the mood when they are ingested [4–6,12]. If the thread of this hypothesis is drawn, we could imagine that the rebalancing of microbiota composition could participate in the maintenance of scalp sebum production but, even more, to the improvement of volunteers' emotions.

## 4. Conclusions

In this study, we presented the hypothesis that psychobiotics, which are molecules having a positive impact on the microbiota bringing benefits to the mental health, can also be topically applied on the skin, due to the similarity between the gut-brain axis and the skin-brain axis.

An active ingredient extracted from *Pogostemon cablin* leaves that is rich in oligosaccharides was evaluated and exhibited interesting biological properties to reduce the production of dry flakes, such as the inhibition of keratinocytes migration that limits the *stratum corneum* thickening, and the restoration of filaggrin and caspase 14 expression leading to a rise in the production of NMFs. At the clinical level, the application of the plant extract normalized the sebum level of the skin and the scalp microbiota composition to one closer to a 'normal' scalp condition, with an increase in the relative abundance of *Staphylococcus* and *Cutibacterium*. These two bacteria are able to produce SFCAs that are essential for the maintenance of the lipid barrier, cell host metabolism and immune system homeostasis. Moreover, *Pogostemon cablin* leaf extract is rich in oligosaccharides that are described to stimulate the production of SCFAs. Knowing that and observing our result with a critical view, we can hypothesize that the active ingredient stimulated sebum production by the skin that led to an enrichment of the relative abundance of *Cutibacterium*, creating a virtuous circle through potential SCFAs production by *Cutibacterium*.

Very interestingly, the normalization of the scalp microbiota composition was associated with a normalization of the skin barrier sebum production and also with a recovery of a normal keratinocyte differentiation level. Regarding the proven effects of SFCAs produced by the gut bacteria to regulate the host cellular metabolism and differentiation and the integrity of the epithelial barrier through the production of mucus, we can hypothesize a similar regulation of skin homeostasis by the skin microbiota acting on both sebum production and keratinocytes differentiation through potential SCFAs release.

The SCFAs are also described to be involved in the differentiation, maturation, genetic regulation and metabolism of microglia that are immune cells acting directly on the mood. Here, an improvement of the mood of volunteers in decreasing dry flakes and in increasing the proportions of bacteria producing SCFAs was demonstrated. We can then hypothesize that the normalization of the scalp microbiota drives the improvement of the mood, and then conferred to the plant extract a psychobiotics effect. As prospects of this study, characterizing the SCFAs production by microorganisms in situ would allow us to answer this hypothesis.

**Author Contributions:** Conceptualization, C.L. and B.S.-P.; methodology, M.M., E.C., C.J., J.B., H.C., C.Z. and A.S.; validation, A.S. and R.R.; formal analysis, M.M., E.C., C.J., J.B., H.C., B.S.-P. and C.Z.; investigation, M.M., E.C., C.J., C.Z. and A.S.; resources, J.T.; writing—original draft preparation, M.M.; writing—review and editing, M.M., E.C., C.J., J.B., H.C., J.T., C.L., B.S.-P., C.Z., A.S. and R.R.; supervision, R.R. All authors have read and agreed to the published version of the manuscript.

**Funding:** This research received no external funding.

**Institutional Review Board Statement:** The study was conducted according to the ethical guidelines of the Declaration of Helsinki. This study, performed on cosmetic products within the definition of article L. 5131-1 of the French Public Health Code, is in accordance with Decree n 2017-884 of 9 May 2017, modifying some regulatory requirements concerning studies involving human participants.

**Informed Consent Statement:** Informed consent was obtained from all participants involved in the study. Written informed consent was obtained from the patients to publish this paper.

**Data Availability Statement:** Data are available on request because of restrictions, e.g., privacy or ethical issues.

**Acknowledgments:** We would like to thank NatExplore (Prouilly, France), Syntivia (Toulouse, France), Qima Life Science (Toulouse, France), Spincontrol (Tours, France), Zurko-CTC (Barcelona, Spain), Dermscan (Villeurbanne, France) and Marina Cavassilas (Paris, France) for their help in the completion of this paper.

**Conflicts of Interest:** The authors declare no conflict of interest. The authors declare that the research was conducted in the absence of any commercial or financial relationships that could be construed as a potential conflict of interest. The funders had no role in the design of the study; in the collection, analyses or interpretation of data; in the writing of the manuscript; or in the decision to publish the results.

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
