# Peer review of "Psychobiotics-like Activity as a Novel Treatment against Dry Scalp Related-White Flakes Production with Pogostemon cablin Leaf Extract"

_cosmetics, doi:10.3390/cosmetics10050130_

Round 1

Reviewer 1 Report

It is a very interesting research,

Author Response

Dear reviewer,

Thank you for your comments, I'm glad to see that you appreciated this manuscript.

Kind regards,

Marie Meunier

Reviewer 2 Report

Psychobiotics-like activity as a novel treatment against dry  scalp related-white flakes production with Pogostemon cablin leaf extract - comments

Comments:

1.         Authors didn’t formulate a clear objective of the work. After the introduction part, Authors shoulg write a clear purpose of the work and describe how it was intended to achieve it.

2.         The passive voice should be used in the manuscript. Instead of "we explored" , "we observed" should be used "it was explored"  , "it was observed" etc. This raises the scientific nature of the work.

3.         In the paper captions and numbering for some tables and figures/photos is missing (page 5 line 202, page 9 line 410, page 9 line 443, page 12 line 555, page 12 line 563, page 13 line 590). It should be completed and the current numbering should be updated.

4.         Page 6 line 295 - authors provided the qualitative composition of the hair shampoo. The quantitative composition of the formulation should also be provided. No data on whether the pH of the resulting formulation was adjusted? What was the pH of the shampoo after manufacture? Was it checked? This is very important.

5.         Page 6 line 295 and page 8 line 394: formulas are the same. Do not repeat the formulation but state it in one place and refer to it.

6.         Page 9 line 410 lack of caption to drawings and lack of reference to them in the text. Throughout the paper, there is mostly a lack of reference to drawings in the descriptions of results. This should be changed.

7.         Page 11 line 524-525 there is no "A" and "B" in the drawings. No indication of "B" in the drawing description, there is only "A".

8.         Page 17 line 698-704 there is lack of labeling of what is "A" and what is "B". The drawings also need to be labeled accordingly.

Author Response

Dear Reviewer,

Thank you for taking the time to read this manuscript. Please find below written in blue my answers to your comments / questions.

1. Authors didn’t formulate a clear objective of the work. After the introduction part, Authors shoulg write a clear purpose of the work and describe how it was intended to achieve it.

The end of the introduction part was reformulated and completed in order to highlight more clearly the purpose of our study.

2. The passive voice should be used in the manuscript. Instead of "we explored" , "we observed" should be used "it was explored"  , "it was observed" etc. This raises the scientific nature of the work.

Some parts of the manuscript were modified with passive voice, especially in results description. You are right, this clearly raises the scientific nature of our works.

3. In the paper captions and numbering for some tables and figures/photos is missing (page 5 line 202, page 9 line 410, page 9 line 443, page 12 line 555, page 12 line 563, page 13 line 590). It should be completed and the current numbering should be updated.

Only the table page 13 line 590 had a numbering and caption but the way to present it is different from figures, with Table number above the table, and legend below. This nomenclature was required by the journal editor. All the missing captions and numbering were completed for the other tables and figures, as requested.

4. Page 6 line 295 - authors provided the qualitative composition of the hair shampoo. The quantitative composition of the formulation should also be provided. No data on whether the pH of the resulting formulation was adjusted? What was the pH of the shampoo after manufacture? Was it checked? This is very important.

pH of the formula wasn't adjusted but remained stable, pH was comprised between 5.5 and 6.0. This detail was added to the manuscript.

5. Page 6 line 295 and page 8 line 394: formulas are the same. Do not repeat the formulation but state it in one place and refer to it.

It has been modified as suggested.

6. Page 9 line 410 lack of caption to drawings and lack of reference to them in the text. Throughout the paper, there is mostly a lack of reference to drawings in the descriptions of results. This should be changed.

More details were added in the materials & methods part to describe the scoring method, and a reminder of the method was also added in the results description.

7. Page 11 line 524-525 there is no "A" and "B" in the drawings. No indication of "B" in the drawing description, there is only "A".

Sorry for that mistake, "B" and "C" were added near the graphics and in the drawing description.

8. Page 17 line 698-704 there is lack of labeling of what is "A" and what is "B". The drawings also need to be labeled accordingly

"A" and "B" labellings were also added in the figure legend.

Reviewer 3 Report

Hi Authors,

The article is interesting and this  study suggests that applying an active ingredient from Pogostemon cablin leaves improves skin health by reducing dry flakes and normalizing sebum production. The observed changes in scalp microbiota composition  lead to the production of SCFAs, potentially influencing skin homeostasis and mood. The findings are promising.

Thank you very much.

Author Response

Dear reviewer,

Thank you for your comments, I'm really glad to see that you appreciated this manuscript.

Kind regards,

Marie Meunier